# Polysaccharides from South Tunisian *Moringa alterniflora* Leaves: Characterization, Cytotoxicity, Antioxidant Activity, and Laser Burn Wound Healing in Rats

**DOI:** 10.3390/plants12020229

**Published:** 2023-01-04

**Authors:** Sameh Sassi Aydi, Samir Aydi, Talel Ben Khadher, Naourez Ktari, Othmane Merah, Jalloul Bouajila

**Affiliations:** 1Laboratory of Biodiversity and Valorisation of Bioresources in Arid Zones (LR18ES36), Faculty of Sciences at the University of Gabes, Gabes 6072, Tunisia; 2Laboratoire de Génie Chimique, Université de Toulouse, CNRS, INP, UPS, F-31062 Toulouse, France; 3Laboratory of Enzyme Engineering and Microbiology, National School of Engineering of Sfax, University of Sfax, Sfax 3038, Tunisia; 4Department of Life Sciences, Faculty of Sciences at the University of Gabes, Gabes 6072, Tunisia; 5Laboratoire de Chimie Agro-Industrielle (LCA), Université de Toulouse, INRAe, INPT, F-31030 Toulouse, France; 6Département Génie Biologique, Université Paul Sabatier, IUT A, F-32000 Auch, France

**Keywords:** biological activities, hydrogel, in vivo activity, polysaccharide characterization, re-epithelialization

## Abstract

Phytochemical properties have recently increased the popularity of plant polysaccharides as wound dressing materials. This work aims at studying the structural characteristics of polysaccharides extracted from *Moringa* leaves (Moringa Leaves Water Soluble Polysaccharide: MLWSP), and its antioxidant activities, cytotoxic effects, and laser burn wound healing effects in rats. This MLWSP was structurally characterized. Results showed 175.21 KDa and 18.6%, respectively, for the molecular weight and the yield of the novel extracted polysaccharide. It is a hetero-polysaccharide containing arabinose, rhamnose, and galactose. XRD suggested a semi-crystalline structure of the studied polymer and FT-IR results revealed a typical polysaccharide structure. It is composed of 50 to 500 µm rocky-shaped units with rough surfaces and it was found to inhibit the proliferation of the human colon (HCT-116) (IC_50_ = 36 ± 2.5 µg/mL), breast (MCF-7) (IC_50_ = 48 ± 3.2), and ovary cancers (IC50 = 24 ± 8.1). The MLWSP showed significant antioxidant effects compared to Trolox (CI_50_ = 0.001 mg/g). Moreover, promising wound healing results were displayed. The effect of MLWSP hydrogel application on laser burn injuries stimulated wound contraction, re-epithelization, and remodeling phases 8 days after treatment. The wound healing potential of MLWSP may be due to its significant antioxidant activity and/or the huge amount of monosaccharide molecules.

## 1. Introduction

The search for new bioactive molecules that can be used in therapy is a major challenge for public health in the treatment of certain diseases [1]. In this context, plants have been screened in several areas to provide new painkillers, antitumor drugs, and antibacterial agents. Chronic wounds represent a major public health problem with more than 6 million of people affected worldwide in the past decade [2].

Wound healing is a complex process affecting a patient’s quality of life [3]. Polysaccharides have been known and exploited for many years due to their abundance, their renewable character, their non-toxicity, and their biodegradability [4]. These biopolymers establish specific interactions with water and can thicken, stabilize, or gel a solution, even at low concentrations [5].

The interest of polysaccharides is not limited to their rheological properties, the biological activities of some of them designate them as molecules capable of modifying the biological functions of certain organisms. Many studies have focused on the isolation and identification of polysaccharides and their biological activities [4,5,6,7]. However, the variety of raw materials and the mechanisms of polysaccharides differ. Recently, several studies reported the wound healing activity of polysaccharides: Zhang et al. [4] showed that the hydrogel has good self-healing and promotes skin tissues, while Kerian et al. [5] revealed that polysaccharides are important wound healing agents. Quan at al. [7] reported that polysaccharides act as bio-multifunctional wound dressings. There are many ways in which polysaccharides contribute to the healing of wounds. They function as antimicrobials, scavengers of free radicals at the wound site, stimulating mitogenic activity, angiogenesis, as well as collagen production [8].

The *Moringa oleifera* tree is part of the monogeneric family Moringaceae and is native to the tropical Northern India. The tree has been cultivated in many parts of the World, including the Mediterranean basin and South Africa [9,10]. It was recently cultivated in Tunisia desert since it resists very well to stressed conditions. It is considered one of the world’s most outstanding nutritional and medicinal trees due to its ability to adapt to different climatic conditions [11,12]. Its leaves, flowers, seeds, pods, and bark have long been used by both Africans and Indians for various purposes such as gas production [9] or as a biostimulant to supplement synthetic fertilizers in agriculture [10].

It is noteworthy that prior works emphasized that *Moringa* leaves were increasingly being incorporated into health foods, but their organic properties have been little studied. Within this framework, Trigo et al. [11] highlighted the abundance of carbohydrates and fibers in the *Moringa* leaves and described their important antioxidant, anti-proliferative, and anti-inflammatory activities without their identification. However, few studies have investigated the polysaccharides derived from *Moringa* leaves.

From this perspective, our research was focused on the investigation of the chemical, physical, and structural characteristics of the water-soluble polysaccharide extracted from *Moringa* leaves (MLWSP) to assess the link between the structure and the bioactivity of these molecules. Cytotoxicity, antioxidant activity, and wound healing ability were also elucidated.

## 2. Results

### 2.1. Structural Analysis of Moringa Leaves Water-Soluble Polysaccharide (MLWSP)

#### 2.1.1. Yield and Composition

Moringa leaves had a polysaccharide content of 18.6%, as shown in Table 1. The MLWSP levels of protein, sugar, and ash were 2.9%, 94.71%, and 3.95%, respectively.

No fats were detected. This showed that MLWSP has a small amount of protein and no fats, while sugars were the most abundant compound. The polysaccharide was thus mostly free of impurities. The estimated average molecular weight of MLWSP was 175.21 KDa (Table 1).

Table 1 showed the physical properties of MLWSP: color and pH. The changes in L*, a*, and b* (lightness, redness, and yellowness) were analyzed. It can be noted that MLWSP offered a light (L* = 31.98), a yellow color (b* = 0.86), and a slightly red color (a* = 0.19). Besides, the pH of a 1% MLWSP solution measured at 25 °C showed an average of 8.28. 

#### 2.1.2. UV–Vis Spectroscopy

MLWSP UV spectra displayed a stronger absorption peak at 200–220 nm, which is due probably to unsaturated carbonyl, carboxyl, etc. Results suggests the presence of polysaccharides. Figure 1 exhibited no optical absorption peaks at 260–280 nm depicting the absence of few proteins or nucleic acids.

#### 2.1.3. X-ray Diffraction Analysis

X-ray diffraction analysis was applied on MLWSP, and the crystalline degree of the polysaccharide was displayed in Figure 2. Results were recorded from 0 to 100 °C. Generally, MLWSP exhibited low crystallinity, suggesting a semi-crystalline structure of the studied polymer. The main crystalline reflections were seen between 5° and 10°, 15°, and at 20° (at the angle 2Θ). That revealed that MLWSP had a classic XRD pattern. It is an amorphous material.

#### 2.1.4. Monosaccharide Composition

Monosaccharides were determined using thin-layer chromatography (TLC) by comparing them with standards arabinose, xylose, fructose, glucose, tagatose, mannose, rhamnose, and galactose (Figure 3A). The analysis showed that the MLWSP was a heteropolysaccharide at the monosaccharide level. Rhamnose, glucose, galactose, and arabinose were identified. 

Additional MLWSP analysis was performed by high-performance liquid chromatography (HPLC) (Figure 3B) to identify monosaccharides. Compared with the monosaccharide standards, MLWSP was principally composed of glucose (14.64%), galactose (14.18%), rhamnose (63%), and arabinose (9.4%) (Table 2). These findings were similar to the TLC results already reported.

#### 2.1.5. FT-IR Spectroscopy 

To further characterize the Moringa leaves polysaccharide, FT-IR spectroscopy was performed.

Figure 4 displays all peaks and bands associated with polysaccharides. All typical peaks and bands associated with polysaccharides were reported. The MLWSP spectrum exhibited eight typical signals of polysaccharides at wavenumbers of approximately 3400, 2920, 2840, 1620,1560, 1400, 1100, and 1000. The broad absorption peak at approximately 3400 cm^−1^ is equivalent to intermolecular or intramolecular induced stretching of hydroxyl (OH) groups, which are always described as a characteristic of polysaccharides.

#### 2.1.6. Scanning Electron Microscopy (SEM)

Figure 5 gives the SEM images of the WSMLP. Our data showed that the MLWSP is composed of 50 to 500 µm geometric shaped units with irregular, rough, and/or rocky surfaces. 

### 2.2. Proprieties and Biological Activities of Moringa Leaves Water-Soluble Polysaccharide (MLWSP)

#### 2.2.1. Functional Proprieties: Water Holding (WHC) and Oil Holding Capacities (OHC)

The results in Table 3 showed that WHC reached 1.54 ± 0.25 mL and WHC attained 1.62 ± 0.17 mL. The water-holding and oil-holding capacities are beneficial properties for the use of these preparations in food, such as sausages and hamburgers. The water-holding capacity of each preparation is important for the juiciness of a final product.

#### 2.2.2. Cytotoxic Activity

The anticancer activity of AWSP with different concentrations was determined by a cell viability assay using human breast MCF-7, colon HCT-116, and ovarian OVCAR cancer cell lines.

Table 4 summarizes the cytotoxic effect of the MLWSP on the three studied cell lines’ viability and (IC_50_) values. The MLWSP was found to inhibit the proliferation of the three studied cancer cells. Results showed that MLWSP exhibited 50% inhibition (IC_50_) for the three cell lines. The overall results indicated that MLWSP exhibited the strongest anticancer activities towards OVCAR cancer cells and the IC_50_ values reported were 24 ± 1.8, 36 ± 2.5, and 48 ± 3.2µg mL^−1^ for OVCAR, MCF-7, and HCT-116 cell lines, respectively.

#### 2.2.3. In Vitro Antioxidant Activity

The results reported in Figure 6 showed the scavenging capacities of the MLWSPs, DPPH, ABTS, and FRAP, compared with the free radical-scavenging activity of BHT.

MLWSP showed better inhibitory activity against ABTS radicals than the DPPH radicals. It has been found that it significantly quenched DPPH (IC_50_ = 1.7 mg·mL^−1^) and ABTS (IC_50_ = 0.082 mg·mL^−1^), and lower inhibitory activity was observed for FRAP activity.

#### 2.2.4. In Vivo Burn Healing Study

The effect of MLWSP on rats

Eight days after wound induction, rats were sacrificed by cervical dislocation and weighed on a balance. The results revealed no significant body weight changes in all groups. Additionally, no death or other undesirable reaction happened throughout the experimental period (data not shown), which reveals that the doses of the MLWSP used for wound healing were nontoxic.

Morphological evaluation

Skin integrity was reestablished, and injured tissue was renewed after laser burning. The MLWSP effect on wound healing was evaluated using the calorimetric assessment. Figure 6 exhibited the progress of treatment and the descriptive collections of wound photos of rats taken on 0, 2, 4, 6, and 8 days after burns from all groups during the wound-healing period. The control group (group I), treated with physiological serum, was compared with groups II, III, and IV, which were treated with glycerol, “Cytol Basic”, and MLWSP gel, respectively. In the beginning of the treatment, laser wounds exhibited an identical red coloration, which evolved to dark brown after one post-operative day and persisted to the third day in all groups. By the fifth day, a novel scab was detected namely in the MLWSP treated wounds, which started to disappear by the seventh day to allow the formation of a new pink-colored epithelium that completely covered the injury, 8 days after laser burn induction (Figure 7). Nevertheless, no complete wound healing was noticed in the other groups by the end of the experiment (8 days).

Wound area assessment

Epithelialization is an essential component of wound healing and is used as a defining parameter of a successful wound closure. Wound areas measurement of different groups of rats are gathered in Table 5. The healing progression was estimated over 8 days by regularly checking the dimension of the wound zone. Data herein showed that all treated groups revealed significant healing effects at the end of the experiment (8 days). Nevertheless, delayed wound healing processes were observed in group I compared to group VI, which presented a significantly quicker closure time. Closure of the wounds seemed to be totally accomplished in the MLWSP-treated group, reaching 0.05 cm against 0.25 cm in “Cytol Basic” group, 0.36 cm in the glycerol group, and 0.89 cm in the physiological serum group (Table 5). 

Hydroxyproline and collagen turnover

Table 6 exhibited a significant increase in hydroxyproline content in “Cytol Centella” treatment (842.82 ± 5.44 mg·g^−1^), namely in MLWSP treated groups (972.54 ± 64.5 mg·g^−1^), when compared to untreated (glycerol and control) ones, inferring more collagen synthesis, re-epithelialization, and fibroblast proliferation. Thus, a faster wound-healing process was observed in treated groups.

Histomorphometric study

Histological observations of wound tissue on the 8th post-laser day revealed full re-epithelialization with a well-structured layer without cell inflammation in the MLWSP treated group (Figure 7). In the same group, a higher presence of skin appendages, such as hair follicles, and sebaceous glands, were observed, which was not achieved in the other groups, showing the tissular re-epithelialization capacity. This result corroborates the complete re-epithelialization capacity of this extract in cutaneous wounds. Equivalent results were observed in Cytol treated group but mild inflammatory cell infiltrations in the perivascular site were seen (Figure 8). However, an invasive inflammatory cell infiltration without an epithelial layer and vacuolization of the dermal cells were seen in the untreated group (Figure 8). 

## 3. Discussion

*Moringa alterniflora* is used worldwide because of its advantages and uses; however, it was recently introduced in the arid zone of Southern Tunisia. To the best of our knowledge, this is the first study reporting the leaves polysaccharide importance in wound healing and their biological activities.

*Moringa alterniflora* has promising phytoconstituents, medicinal properties, and nutritional benefits [13]. Several African and Asian countries use it as a source of food or as a raw material [14]. Recently, researchers have been highly interested in polysaccharide extracts from natural medicines because of their structural diversity and low toxicity [15,16,17]. Herein, polysaccharides isolated and purified from leaves of *Moringa alterniflora* cultivated in the arid regions of southeastern Tunisia exhibited 18.6% of the dry material. This value was superior to another polysaccharide content, such as in Bergenia (15.8%) [18], and Jackfruit (11.8%) [19]. The proportion of water to raw material, time, and temperature might all have an impact on these variations [20]. Extraction methods may influence the polysaccharide yield of *Moringa* leaves: Microwave-assisted extraction confirmed a lower yield, which did now not exceed 3% [21], while hot water extraction followed by ethanol precipitation revealed 6.8% [22], and coupling ionic liquid separation gadget with ultrasound irradiation exhibited a better yield attaining to 75% [23]. In MLWSP, sugars were the most abundant detail, as described in the maximum extracted polysaccharides [24]. In addition to sugars, Kolsiet al. [15,16] found that polysaccharides are commonly linked to sulfate radicals and proteins. The molecular weight of polysaccharides varies considerably; as suggested by several authors, polysaccharides have extensively unique common molecular weights [25,26,27]. Changes in molecular weight have a significant impact on the polysaccharide’s rheological and viscoelastic properties [28]. Figure 1 exhibited no optical absorption peaks at 260–280 nm, depicting the absence of proteins or nucleic acids, this corroborates previous results reported by Zhou et al. [29]. X-ray diffraction showed the predominance of the amorphous components of MLWSP. Additional data reported the amorphous nature of water-soluble polysaccharides [13,15]. As suggested by many authors, polysaccharides crystallize only into tiny crystals [30,31,32,33]. X-ray diffraction suggested non- or semi-crystalline polymers [31]. This crystallinity is very common in polysaccharides of nature [32,33]. Our data agrees with that of the water-soluble polysaccharides of potato peels [34], but are different from data of *Spirulina* polysaccharides [35]. Thin-layer chromatography (TLC) was widely described as a common approach for routine examination of monosaccharides or polymers in pharmaceutical and herbal medication samples [20]. We discovered that water-soluble polysaccharidea extracted from *Moringa* leaves consisted of monosaccharides such as glucose, galactose, rhamnose, and arabinose sugar composition. Our previous work on *spirulina* polysaccharide reported that rhamnose was the major sugar in *Spirulina* [35]. This was also reported by Kolsi et al. [16] in seaweed.

The precise chemical structure of polysaccharides is clarified using a variety of analytical methods, including spectroscopic, chemical, and chromatographic investigation [36]. Infrared (IR) spectroscopy is a well-known technique that has been widely applied in polysaccharide structural studies. The well-divided absorption regions of characteristic groups in the FTIR spectrum can make interpretation easier and serve as a good guide for identifying carbohydrates [37]. The typical bands of polysaccharides were reported by Borovkova et al. [37]

The strong absorption band between 3200 and 3600 cm with a peak at 3400 cm^−1^ results from the stretch vibration of O–H in polysaccharides. The observed peaks at 2920 cm^−1^ and 2840 cm^−1^ were probably due to CH_2_ stretching and bending vibrations of free sugars [38]. However, the C–OO symmetric stretching vibration takes place within the wavelength from 1600 cm^−1^ [39]. CH_2_ and the numerous C-OH deformations encountered in carbohydrates at 1400 cm^−1^ [37]. Peaks at 1100 cm^−1^ and 1000 cm^−1^ could be ascribed to the C–O stretching vibration in the pyranose ring structure of C–O–H [36] and may be related to the presence of uronic acid. Besides, Tang et al. [40] pointed out that the signal at 1240 cm^−1^ might be related to the asymmetrical stretching vibration of S–O, which is evidence of the sulfated group. These findings showed that our polysaccharide is a heteropolysaccharide containing uronic acid, a sulfated group, and α- and β-glycosidic bonds.

SEM should be a great technique for examining the surface structure of polysaccharides because to its high resolution, expansive field of view, and stereoscopic capabilities [41]. Data herein showed that MLWSP was composed of irregular and rough particles. This could be the prerequisite for gaining insight into structure–activity relationship [38]. Different structures that demonstrated different polysaccharide architectures were provided in the literature [22,38]. This can be mostly because of how the sample was prepared before microscopy.

Polysaccharides extracted from *Moringa* leaves exhibited valuable effects against colon, breast, and ovarian cancer cell lines. It has been shown that polysaccharides with a clear antitumor impact can influence the development of tumors in a variety of cell lines, mostly by preventing the growth of tumors, causing apoptosis, boosting immune response, and coordinating chemotherapy treatments [42,43]. A polysaccharide produced by the Solanaceae plant *Lyciumbarbarum* stopped the development of MCF-7 tumor cells and caused their apoptosis [44]. Hence, other natural sources should be explored for their anticancer potential.

Additional significant bioactivities that might be regarded as typical for many plant polysaccharides are their anti-oxidative actions [16,24,34,45]. We supported these findings in our investigation. Numerous studies have shown that polysaccharides have antioxidant activity, reducing power, and may scavenge radicals such as 2,2-diphenyl-1-picrylhydrazyl (DPPH), hydroxyl, peroxyl, alkyl, H_2_O_2_, superoxide, and ABTS [16,20,35,46].

Indeed, several studies reported that the antioxidant activities of polysaccharides extracted from numerous medicinal plants were significant and that those carbohydrates may be used as potential antioxidants [15,35]. Similar data were recently reported [45,47]. Antioxidant capacity showed a significant and positive correlation with polyphenols contents, as most of these constituents are proven scavengers of free radicals [48]. These qualities can be considered and perhaps used in the medical or food industries. Overall, these results demonstrated that MLWSP exhibited a higher wound healing potential due to its high antioxidant activities in vitro.

Concerning the study of wound healing, natural bioactive polysaccharides have gained increased interest for medical treatments, most notably tissue engineering. Modern wound dressings are increasingly using polysaccharide biopolymers, which have been shown to have beneficial effects on human skin cells [49]. In this study, the appearance of the wounds in the four groups of rats was constantly evaluated for 8 days to assess the progress of wound healing. Photographs of the wound may be a crucial indicator of how well the wound is healing [50,51]. By decreasing downtime and accelerating re-epithelialization, MLWSP gel seems to be perfect for wound healing. This could be allocated to its important antioxidant activity [52]. This finding is in accordance with previous work by Ktari et al. [53] and Mapoung et al. [54] showing the benefic effect of polysaccharides in healing wounds. The amino acid hydroxyproline is normally found in significant quantity in collagen and its measurement can be used as an indicator of collagen formation [55,56]. Data revealed that MLWSP significantly accelerated wound healing after 8 days of treatment, attested by higher wound area measurement and a higher content of collagen (estimated by hydroxyproline content), and was confirmed by histological examination, when compared with control, cytol centella^®^, physiological serum, and glycerol groups. In fact, the increasing collagen synthesis and deposition displayed a critical role in wound closure [56]. In this trend, Trablesi et al. [57] have previously reported that the high concentration of hydroxyproline can be explained by the capacity of polysaccharides to promote fibroblast migration by supplying a connective tissue matrix. In contrast, the rats given MLWSP treatment showed complete wound site size reduction in less time than the control group, as well as complete restoration of the original skin structures. Similarly, the wound contraction percentage increased cell proliferation and epithelization while reducing infections [53]. The physiological serum-treated group, in comparison, had an open wound that healed more slowly than the other groups. The most effective way for assessing the extent of wound healing is a histological examination of wound tissues [56]. Hematoxylin-colored micrograph slices are examined under light microscopy to evaluate the inflammatory and regenerative features, such as tissue congestion and neutrophil infiltration. Indeed, the ability of MLWSP to speed up wound healing may be attributed to a variety of mechanisms, including an increase in the rate of re-epithelialization and neovascularization, the capacity of harmful free radicals, the reduction of inflammation, and the control of infection, which may be attributed to the antioxidant activity of polysaccharide constituents.

## 4. Materials and Methods

### 4.1. Material

*M. alterniflora* was cultivated in south Tunisian fields (Tataouine). After collecting the yield, leaves were carefully washed with tap water and rinsed with distilled water, then spread air-dried under shade at room temperature for 2 weeks. Dried leaves were ground into a powder with mortar and pestle and stored in a desiccator for further use. Voucher specimens of *Moringa* leaves were identified by professor Lotfi Abdallah from the Department of Life Sciences, Faculty of Sciences, at the University of Gabes, and kept at the herbarium of the Laboratory of Biodiversity and Valorization of Bioresources in Arid Zones (Faculty of Sciences, University of Gabes, Gabes, Tunisia) with the reference LBVAZ-02/019.

### 4.2. Extraction of MLWSP

*Moringa* leaves water-soluble polysaccharide (MLWSP) was assessed by the method described by Mapounget al. [54]. Briefly, *Moringa* leaves powder was pre-extracted with 95% ethanol at room temperature (23 ± 3 °C), to eliminate pigments. The dry residue was extracted twice with 20 vols of deionized water at 90 °C for 4 h, where the suspension was continuously mixed using a magnetic agitator (AREX Velp-Scientifica, Usmate, Italy). Extracts were combined and filtered, and filtrates were then evaporated using a rotary vacuum evaporator (Büchi Rotavapor R-200 (Büchi Labortechnik, Flawil, Switzerland). The concentrated liquid was precipitated with 95% (v v^−1^) ethanol at 4 °C for 24 h and then centrifuged (4500 rpm) for 15 min using a HERMLE Z 513 K centrifuge (HERMLE Labortechnik, Wehingen, Germany). The final precipitate was redissolved in double distilled water. The water phase was dialyzed at 4 °C against distilled water for 2 days. The dialysate was concentrated by the Büchi Rotavapor under reduced pressure and freeze-dried using a freeze dryer (Bioblock Scientific Christ ALPHA 1–2, IllKrich-Graffenstaden-Cedex, France) to obtain MLWSP, which was then stored at −20 °C for further use.

### 4.3. Physico-Chemical Analysis

Color, pH (1% solution at 25 °C), and viscosity at various concentrations of MLWSP in H_2_O (0.5, 1, and 1.5 g L^−1^) were determined. The color was evaluated using a Color Flex spectrocolorimeter (Hunter Associates Laboratory Inc., Reston, VA, USA) and was reported as L*, a*, and b* values, referring to the measuring parameters of lightness, redness, and yellowness, respectively. The sample was filled in a 64 mm glass cup with three readings. The latter was determined in triplicate. The white tile and black glass were used to standardize the equipment. The pH was measured using a digital pH meter (Mettler-Toledo AG, Schwerzenbach, Switzerland). Viscosity was determined at 25 °C by using a digital viscometer (NDJ-1, Japon) at 30 rpm spindle rotation. The moisture and ash content were determined using the AOAC [58] method. Total sugars were determined by the phenol-sulfuric acid method [59]. Crude fat was determined gravimetrically by Soxhlet extraction of dried samples. Crude protein content was estimated by multiplying total nitrogen content by the factor of 6.25.

### 4.4. Thin Layer Chromatography (TLC)

Different standards were used for this study. Polysaccharides may contain galactose, arabinose, rhamnose, tagatose, glucose, xylose, mannose, and fructose. Four Silica gel plates (type Silica gel 60 F 25) were the chromatographic plaques used, with a thickness of 0.25 mm. The mobile phase used was Butanol, Acetic acid, and Water, in the proportions of 2:1:1 (v v^−1^). Fluorescent spots were located under a 254 nm UV lamp.

### 4.5. Spectroscopic Analysis

#### 4.5.1. UV Absorption Peak Detection

MLWSP was dissolved in distilled water to a final concentration of 0.1%. The UV absorption spectrum of the sample was recorded in the wavelength range of 200–800 nm [60] using a UV-VIS Spectrophotometer (2005, JP Selecta S.A., Barcelona, Spain).

#### 4.5.2. FT-IR Spectrometric Analysis

FT-IR spectrum of MLWSP was determined on a Nicolet FTIR spectrometer equipped with horizontal attenuated total reflection (ATR) accessory. The internal crystal reflection was made from zinc selenide and had a 45° angle of incidence to the IR beam. The spectrum was acquired at 4 cm^−1^ resolution, and the measurement range was 500–4000 cm^−1^ (mid-IR region) at room temperature. The spectral data were analyzed by the OPUS 3.0 data collection software program (Bruker, Ettlingen, Germany).

#### 4.5.3. X-ray Diffraction (XRD)

The XRD pattern of MLWSP was recorded at room temperature on an X-ray diffractometer (D8 advance, Bruker, Germany). The data were collected in the 2Θ ranges 5–80° with a step size of 0.05° and accounting time of 5 s/step.

#### 4.5.4. HPLC Analysis

An aliquot of 2 mg of MLWSP was hydrolyzed in 250 µL of 2 M sulfuric acid (H_2_SO_4_) at 100 °C for1 h. A 20 µL hydrolysate was added to 980 µL of deionized water and filtered through anhydrophobic PTFE 0.45 µm membrane filter (Sartorius GmbH, Goettingen, Germany). Monosaccharide composition was analyzed by HPLC using a sugar KS-800 column with a mobile phase of 0.001 M NaOH, a flow rate of 0.5 mL min^−1^, and a column temperature of 50 °C. The monosaccharide composition assays were performed in two independent experiments. Glucose, fructose, sucrose, gluconic acid, mannose, arabinose, galactose, and xylose were used as standard monosaccharides.

### 4.6. Scanning Electron Microscopic (SEM)

Polysaccharides were examined by scanning electron microscopy (SEM) model JEOL (JSM-IT100). The dried polysaccharide was mounted on a metal stub and sputtered in carbon conductive adhesive tapes and the images were observed. The accelerating voltage was 1 kV.

### 4.7. Water-Holding and Oil-Holding Capacities (WHC and OHC)

WHC and OHC were assayed by the method of Nguyen et al. [61]. Briefly, 0.5 g of MLWSP was dissolved in 50 mL of distilled water or 10 mL of Corn oil. The mixed solution was kept at room temperature for 1 h and then centrifuged at 8000 rpm for 20 min. The supernatant was removed prudently, and the centrifuged tube was kept on a filter paper for 30 min to drain, after being oriented to a 45° angle. The ratio between the weight of the tube content after draining and the weight of the MLWSP was determined and the capacity (%) was reported as g of water or oil bound per g of the MLWSP on a dry basis.

### 4.8. Biological Activities Evaluation

#### 4.8.1. In Vitro Antioxidant Assays

##### DPPH Radical Scavenging Assay

The stable free radical scavenging activity was evaluated using the DPPH assay [62]. MLWSP was mixed, at equal volume, with 2,2-diphenyl-1-picrylhydrazyl (DPPH, 100 mM), then incubated at room temperature for 15 min. The absorbance was determined at 517 nm. Butylated hydroxy-toluene (BHT) was used as a positive control. The 50% inhibitory concentration (IC_50_) was expressed as the quantity of the extracts to react with a half of DPPH radicals. The IC_50_ values were calculated using the linear regression analysis and used to indicate antioxidant capacity.

##### Ferric Reducing Antioxidant Power Assay (FRAP)

The FRAP assay was done according to the process of Abreu et al. [63]. The FRAP reagent was prepared from 0.3 M acetate buffer (pH 3.6), 10 mmol 2,4,6-tripyridyl-s-triazine (TPTZ) solution in 40 mmol hydrochloric acid and 20 mmol iron (III) chloride solution with the ratio 10:1:1 (v v^−1^). Briefly, 50 μL of MLWSP sample (three replicates) were added to 1.5 mL of the FRAP reagent. Four min after, the absorbance was determined at 593 nm. The standard curve was constructed using FeSO4solution (from 50 to 200 μg mL^−1^).

##### ABTS Assay

The radical scavenging activity of MLWSP was determined according to the procedure of Dissanayak et al. [64] with slight modifications. 2.2-azino-bis3-ethylbenzothiazoline-6-sulfonic acid (ABTS) (7 mM) and potassium persulfate (2.45 mM) solutions were mixed and stored in a dark room for 12–16 h before to use. Before the analysis, the ABTS solution was diluted with ethanol to an absorbance of 0.700 ± 0.05 at 734 nm. Following the addition of 4.5 mL of the ABTS reaction mixture to the various concentrations (50–250 µgmL^−1^) of the MLWSP (1 mg mL^−1^), the reaction mixture was vortexed. After keeping at room temperature for 15 min, the absorbance of the samples was read at 734 nm. The results were assessed as IC_50_ values.

#### 4.8.2. Cytotoxic Activity

Cytotoxicity of MLWSP against human breast (MCF-7), ovarian (OVCAR), and colon (HCT-116) cancer cell lines was estimated by the 3-(4,5-dimethylthiazol-2-yl)-2,5-diphenyltetrazolium bromide (MTT) assay of Ben Khadher et al. [60]. Briefly, 100 µL of cells were distributed in 96-well plates at a concentration of 104 cells per wells and incubated at 37 °C during 24 h. Then, 100 µL of cells in exponential growth phase were incubated in a fully humidified atmosphere at 37 °C for 48 h with the addition of 100 µL of the culture medium, supplemented with 2 mM L-glutamine and 50 μg mL^−1^ gentamycin, containing MLWSP at a concentration of 50 mg L^−1^. The medium was removed, and cells were treated with MTT solution 50 µL, 1 mg mL^−1^ in phosphate buffered saline (PBS) and incubated at 37 °C for 40 min. MTT solution was then discarded and dimethyl sulfoxide (DMSO) (50 µL) was added to dissolve in soluble blue crystals. Optical density was determined at 605 nm. Tamoxifen was used as a positive control.

### 4.9. In Vivo Study of the Effect of MLWSP on Laser Wound Healing

#### 4.9.1. In Vivo Assay

Adult Wistar rats (190–200 g) were gently provided by the Department of Life Sciences, Faculty of Sciences, at the University of Gabes, Tunisia. The animals were caged under controlled conditions of light (12 h light/dark cycles), room temperature (23 ± 1 °C), and relative humidity (50% ± 10%) with free access to food and water ad libitum. The general guidelines on the use of living animals in scientific investigations (Council of European Communities, 1986) and the guidelines for the care and use of laboratory animals controlled by the Tunisian Research Ministry were followed. Animals were treated in the respect of ethic and deontology and all the procedure was accorded with Guidelines for Ethical Conduct in the Care and Use of Laboratory Animals [65].

#### 4.9.2. Wound Healing Activity

##### Fractional CO_2_ Laser Burn Creation

Eighteen rats were anesthetized with ketamine, 50 mg/kg, along with 5 mg/kg of midazolam. The back of each animal was shaved and exposed to partial-thickness skin burns (wound area = 2 cm^2^) by a CO_2_ Fractional Laser System (DSE, Seoul South Korea) as follows: density: (level = 20, line = 29 × 29, dot = 0841), energy level = 25 MJ, and depth level = 4.

##### Experiment Protocol

After CO_2_ laser burns, the animals were divided into three groups (n = 6) and treated with glycerol solution (30%), “Cytol centella”, and MLWSP hydrogel, respectively. Each animal was housed separately and treated every day until the first group was completely healed. “Cytol centella” cream was purchased from local pharmaceutical industry. The hydrogel was prepared by dissolving the lyophilized MLWSP in glycerol solution (30%), to give a final concentration of 15 mg/mL. The mixture was kept under agitation until a transparent hydrogel was formed.

##### Hydroxyproline Level Measurement

Hydroxyproline level was assessed according to Lodhi et al. [66]. Hydroxyproline concentrations were then calculated from the linear standard curve and presented as mg/100 mg of dry tissue weight.

##### Histopathological Examinations

The skin and sub-plantar muscle samples were fixed in 10% neutral buffered formalin solution, embedded in paraffin wax, and cut into 5 mm thick sections on a sliding microtome (Leica, Wetzlar, Germany). The specimens were deparaffinized with 80% xylene and ethyl alcohol, rinsed with phosphate buffer solution (PBS, pH 7.4), and stained with Mayer’s hematoxylin solution and 1% eosin alcohol solution. Finally, a light microscope (Olympus, Tokyo, Japan) was used to examine the serial sections to identify the morphology of skin tissues.

### 4.10. Statistical Analysis

All experiments were done in triplicate, and data were expressed as mean value ± standard deviation. The statistical analyzes were performed using the one-way analysis of variance (ANOVA) procedure with the Statistical Package for the Social Sciences (SPSS) version 20.0 software (IBM Corp. 2011, Armonk, NY, USA). When *p* < 0.05, differences were considered as statistically significant according to Fisher’s LSD test.

## 5. Conclusions

According to this study, *Moringa* leaves provided a novel polysaccharide (MLWSP). It had a molecular weight of 175.21 KDa, and glucose, galactose, arabinose, and rhamnose made up most of its composition. Notably, MLWSP’s distinct structure suggested that it may be widely used in the food industry. It exhibited advantageous bioactivities that might likely be responsible for wound closure and re-epithelialization and markedly enhanced the rate of hydroxyproline in the wound site. These properties imply MLWSP may represent a novel biomaterial for therapeutic processes in applications for wound healing. Nevertheless, numerous innovative composite hydrogel dressings are still in the laboratory research stage and have not yet been extensively utilized because of different limitations like cost, safety, and the production process. Future research on hydrogels and their dressings must go deeper; enhancing their mechanical characteristics and ability to promote healing is still the main objective.

## Figures and Tables

**Figure 1 plants-12-00229-f001:**
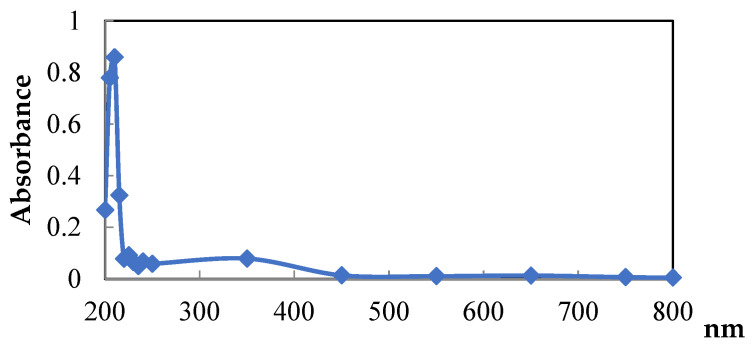
Scan of SWSP within the wavelength range of 200–800 nm.

**Figure 2 plants-12-00229-f002:**
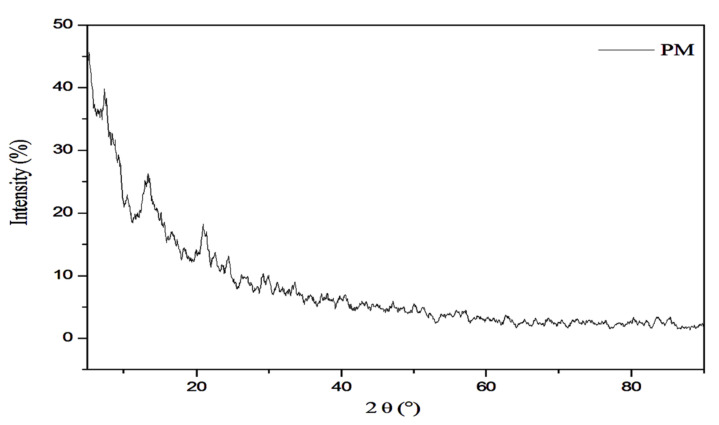
X-ray diffraction patterns of MLWSP.

**Figure 3 plants-12-00229-f003:**
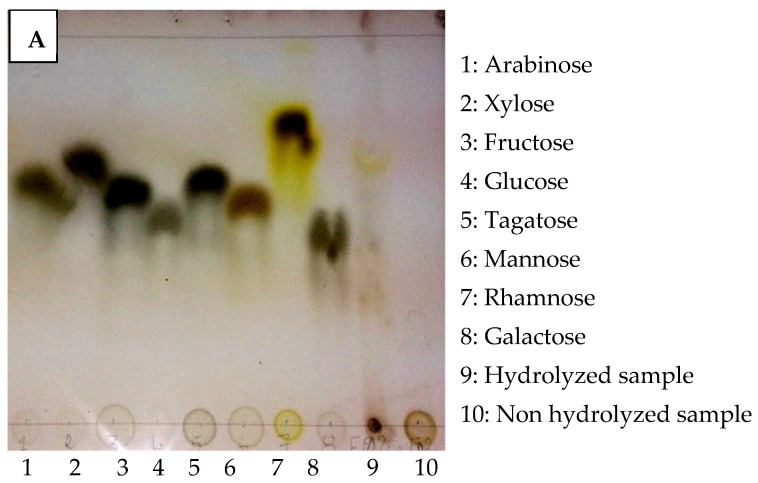
(**A**) TLC profile of the MLWSP 1: Arabinose, 2: Xylose, 3: Fructose, 4: Glucose, 5: Tagatose, 6: Mannose, 7: Rhamnose, 8: Galactose, 9: Hydrolyzed sample, 10: Non hydrolyzed sample. (**B**) HPLC chromatogram profiles of MLWSP. Samples were applied to the Sugar KS-800 column at a flow rate of 0.5 mL/min.

**Figure 4 plants-12-00229-f004:**
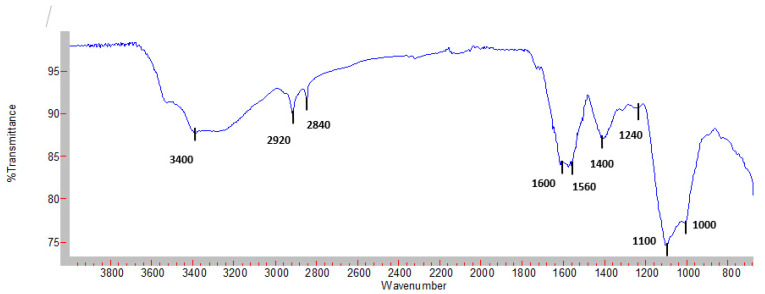
Fourier transformed the infrared spectrum of MLWSP.

**Figure 5 plants-12-00229-f005:**
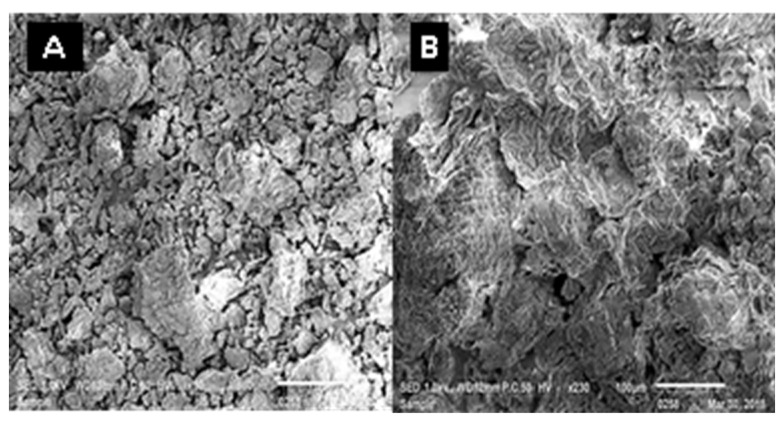
SEM images of MLWSP. (**A**) Morphology at 50× (scale bar is 500 µm). (**B**) Morphology at 250× (scale bar is 100 µm).

**Figure 6 plants-12-00229-f006:**
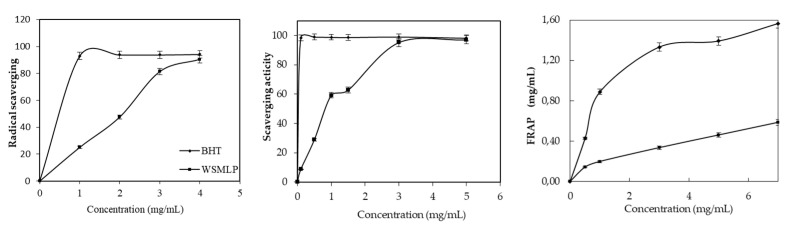
Antioxidant activity assessed by the (DPPH) radical scavenging capacity assay, (ABTS) assay, and ferric reducing (FRAP) assay in MLWSP. Values are the mean of three replicates ± standard deviation.

**Figure 7 plants-12-00229-f007:**
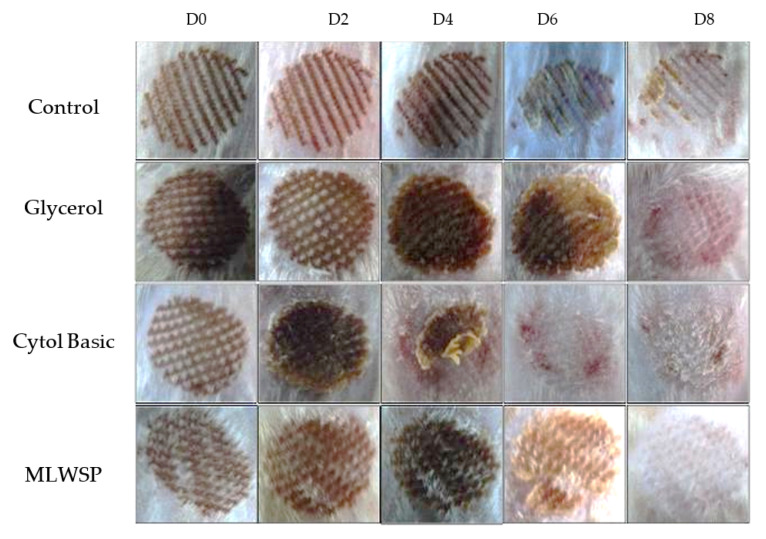
Laser Burn wounds chronicity taken for the different groups on days 0, 2, 4, 6, and 8. Group I was treated with physiological serum; Group II was treated with glycerol; Group III was treated with “Cytol Basic”, and Group IV: was treated with MLWSP hydrogel.

**Figure 8 plants-12-00229-f008:**
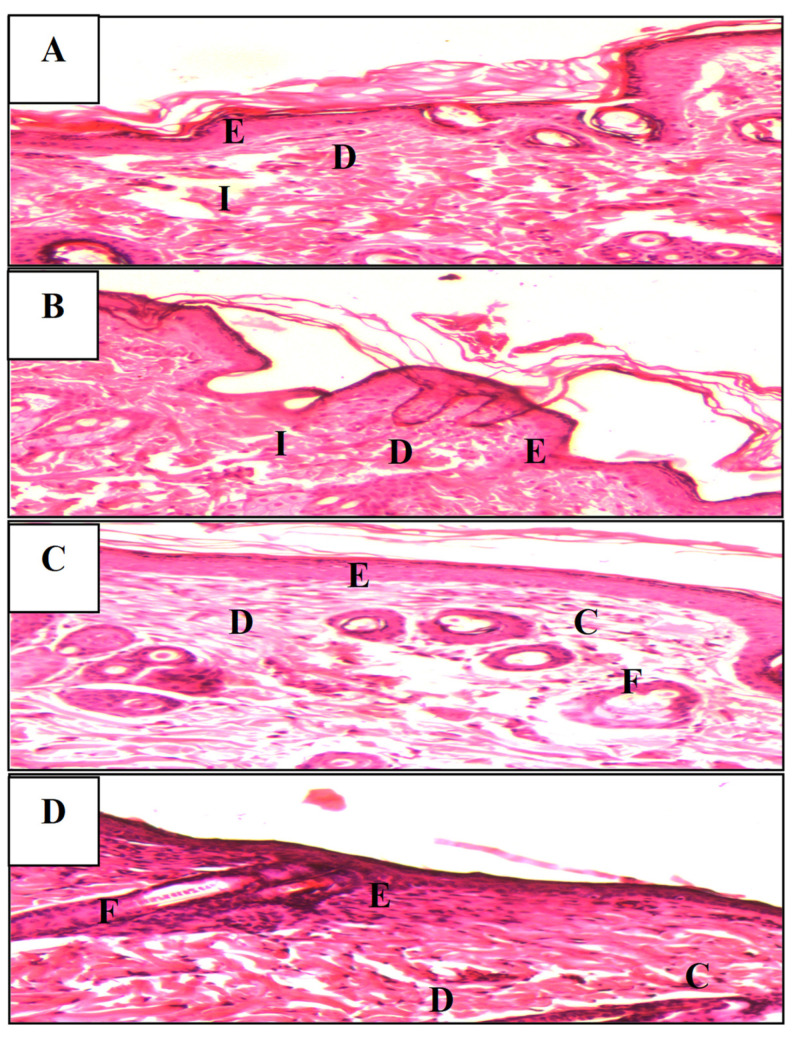
Representative photomicrographs of the epidermal and dermal architecture of wounds on the 8th day of treated rats with (**A**): physiological serum (Group I); (**B**): glycerol solution (Group II); (**C**): Cytol Centella (Group III); or (**D**): MLWSP hydrogel (Group IV). Tissues were stained with hematoxylin-eosin and visualized at 100× magnification. E: epidermis; D: dermis; F, hair follicle; C, collagen; and I, inflammatory cell.

**Table 1 plants-12-00229-t001:** Yield, chemical composition, and physical properties of MLWSP.

	Composition (g/100 g)
Yield	18.6 ± 0.42
Protein	2.9 ± 0.24
Fat	0.00 ± 0.00
Ash	3.95 ± 0.08
Total sugars	94.71 ± 0.21
pH (solution 1%)	6.76
Color	
L*	48.68 ± 1.1
a*	−1.24 ± 0.01
b*	8.18 ± 0.85
Molecular weight (KDa)	175.21

Values are given as mean ± SD (n = 3); SD: standard deviation.

**Table 2 plants-12-00229-t002:** Monosacharide composition of the MLWSP.

Monosaccharides	Glucose	Galactose	Rhamnose	Arabinose
%	14.64	14.18	63	9.4

**Table 3 plants-12-00229-t003:** Water-holding and oil-holding capacities of MLWSP.

Properties	Capacities (%) *
Water-holding capacity	1.54 ± 0.25
Oil-holding capacity	1.62 ± 0.17

* Values are given as mean ± SD (n = 6); SD: standard deviation.

**Table 4 plants-12-00229-t004:** Cytotoxic effects of MLWSP on MCF-7, HCT-116, and OVCAR cell lines IC_50_ values.

Celllines	MLWSP	Tamoxifen
MCF-7	* 48 ± 3.2	0.15 ± 0.02
HCT-116OVCAR	* 36 ± 2.5 * 24 ± 1.8	0.14 ± 0.02 0.19 ± 0.03

* Values are given as mean ± SD (n = 3); SD: standard deviation.

**Table 5 plants-12-00229-t005:** Wound areas measurement of different groups of rats.

Group	Days				
	0	2	4	6	8
Group I	1.60 ± 0.22 a	1.56 ± 0.28 a	1.52 ± 0.14 b	1.22 ± 0.11 b	0.89 ± 0.17 d
Group II	1.60 ± 0.13 a	1.51 ± 0.11 a	1.43 ± 0.10 ab	0.95 ± 0.15 a	0.36 ± 0.02 c
Group III	1.61 ± 0.35 a	1.42 ± 0.36 a	1.23 ± 0.19 a	0.87 ± 0.08 a	0.25 ± 0.03 b
Group IV	1.62 ± 0.18 a	1.39 ± 0.32 a	1.21 ± 0.29 a	0.67 ± 0.05 a	0.01 ± 0.001 a

Values are given as mean ± SD (n = 5 rats per group). a, b, c, d: different letters for each column represent significant differences at *p* < 0.05. Unit is cm. Group I treated with physiological serum; Group II treated with glycerol; Group III treated with “Cytol Basic”, and Group IV treated with MLWSP.

**Table 6 plants-12-00229-t006:** Hydroxyproline amounts in different experimental animal group.

Groups	Hydroxyproline (mg/g of Tissue)
Control	642.88 ± 48.9 d
Glycerol	735.16 ± 43.3 c
Cytol	842.82 ± 51.4 b
MLWSP	972.54 ± 64.5 a

Values are given as mean ± SD (n = 3). a, b, c, d: different letters represent significant differences at *p* < 0.05.

## Data Availability

Not applicable.

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
