# Peer review of "Polysaccharides from South Tunisian Moringa alterniflora Leaves: Characterization, Cytotoxicity, Antioxidant Activity, and Laser Burn Wound Healing in Rats"

_plants, 2023, doi:10.3390/plants12020229_

Round 1
Reviewer 1 Report
These are my main comments on the manuscript (plants-2125977) entitled “Polysaccharide from South Tunisian Moringa alterniflora. Lam leaves: Characterization, cytotoxicity, antioxidant activity and laser burn wound healing in rats”. Following substantial revisions should be incorporated in the manuscript prior to acceptance.
1. I have concerns about the manuscript sections that I believe need to be addressed in order to improve its clarity.
2. A hypothesis for this work is needed.
3. In results, Statistical methods are missing in methods section. The authors must present the values (F-values, degree freedom, p-value, etc.) obtained in each statistical analysis.
4. Other revisions could be checked in PDF attached.

Author Response
Dear reviewer
We appreciate the time and effort that you and the reviewers dedicated to providing feedback on our manuscript and are grateful for the insightful comments on and valuable improvements to our paper. We have incorporated most of the suggestions made by the reviewers. Those changes are marked up using the “TrackChanges” function within the manuscript. Please see below, in green, for a point-by-point response to the reviewers’ comments and concerns.
We hope that, in the current form the MS can be accepted for publication in Plants.
Looking forward to hearing from you.
Sincerely,
Prof. Othmane Merah
- I have concerns about the manuscript sections that I believe need to be addressed in order to improve its clarity.
Authors would like to thank the reviewer for all valuable comments and remarks. All suggestions were changed as recommended.
- A hypothesis for this work is needed.
Corrected, the hypothesis of this work was added in the last paragraph of the introduction section.
- In results, Statistical methods are missing in methods section. The authors must present the values (F-values, degree freedom, p-value, etc.) obtained in each statistical analysis.
Corrected, statistical method was described in the statistical analysis section.
- Other revisions could be checked in PDF attached.
Many thanks, all suggestions were changed as suggested.
Reviewer 2 Report
Reviewer comments and suggestions
The study aimed to study the structural characteristics of polysaccharides extracted from Moringa leaves (Moringa Leaves Water Soluble Polysaccharide: MLWSP), and their antioxidant activities, cytotoxic effects, and laser burn wound healing in rats.
The study results suggested that 175.21 KDa and 18.6% respectively for the molecular weight and the yield of the novel extracted polysaccharide.
XRD suggested a semi-crystalline structure of the studied polymer and FT-IR results revealed a typical polysaccharide structure. The study confirms that polysaccharide was found to inhibit proliferation of human colon (HCT-116) (IC50= 36±2.5 µg/ml), breast (MCF-7) (IC50= 48±3.2) and ovary cancers (IC50= 24±8.1). The effect of MLWSP hydrogel application on laser burn injuries stimulates wound contraction, the reepithelization and remodeling phases, eight days after treatment.
Overall, the manuscript was well written. However, a few concerns/comments needed to be explained/modified.
- Line number 50-51, the author said that many studies but only cited one. Similar comments for please check lines 344 -346
- Line 53 please explore these studies (4,5,7)
- Lines 62-63 explore the studies so that the common reader could also get knowledge about it
- Typo error in line 66 cytotoxicity
- Discussion first para the first para needs to be discussed the novelty of this study.
- Line 296-297 is this your result, if yes then why did the authors add reference 29 there
- Spirulina should be italics please modify lines 303, 308
- Line 323-324 is not a good choice to present like this (40), please change with the author's name et al.
- Line 382 Please mention the outcome of your study here
- Check section 4.8 please discuss the points completely
- Similar references could also be cited Recent developments in Moringa oleifera Lam. polysaccharides: A review of the relationship between extraction methods, structural characteristics, and functional activities
https://en.cnki.com.cn/Article_en/CJFDTOTAL-SPYK201314009.htm
12. All references need to be changed, check the author's name and how they were cited in MDPI
Author Response
Dear Reviewer
We appreciate the time and effort that you and the reviewers dedicated to providing feedback on our manuscript and are grateful for the insightful comments on and valuable improvements to our paper. We have incorporated most of the suggestions made by the reviewers. Those changes are marked up using the “TrackChanges” function within the manuscript. Please see below, in green, for a point-by-point response to the reviewers’ comments and concerns.
We hope that, in the current form the MS can be accepted for publication in Plants.
Looking forward to hearing from you.
Sincerely,
Prof. Othmane Merah
Overall, the manuscript was well written. However, a few concerns/comments needed to be explained/modified.
- Line number 50-51, the author said that many studies but only cited one. Similar comments for please check lines 344 -346
Corrected as recommended.
- Line 53 please explore these studies (4,5,7)
Corrected, the following paragraph was added:
"Recently, several studies reported the wound healing activity of polysaccharides: Zhang et al. [4] showed that the hydrogel has good self-healing and promotes skin tissues, while Kerian et al. [5] revealed that polysaccharides are important wound healing agents. Quan at al. [7] reported that polysaccharides acts as bio-multifunctional wound dressings"
- Lines 62-63 explore the studies so that the common reader could also get knowledge about it
Corrected, this sentence was added: "such as gas production [9] or a biostimulant to supplement synthetic fertilizers in agriculture".
- Typo error in line 66 cytotoxicity
Corrected
- Discussion first para the first para needs to be discussed the novelty of this study.
Corrected, the following sentence was added:
"Moringa alterniflora was used worldwide because of its several advantages and uses, however, it was recently introduced in the arid zone of southern Tunisia. To the best of our knowledge, this is the first study reporting the leaves polysaccharide importance in wound healing and their biological activities."
- Line 296-297 is this your result, if yes then why did the authors add reference 29 there
Agreed, this was corrected by adding following sentence " this corroborates previous results reported by Zhou et al."
- Spirulina should be italics please modify lines 303, 308
Corrected
- Line 323-324 is not a good choice to present like this (40), please change with the author's name et al.
Corrected, the name of the author's "Tang et al." was added
- Line 382 Please mention the outcome of your study here
Corrected, actually, this paragraph mention the outcome of our study. However, since it corroborates the study of Eleroui et al. [56].
- Check section 4.8 please discuss the points completely
Corrected, detailed protocols added
- Similar references could also be cited Recent developments in Moringa oleifera Lam. polysaccharides: A review of the relationship between extraction methods, structural characteristics, and functional activitieshttps://en.cnki.com.cn/Article_en/CJFDTOTAL-SPYK201314009.htm
Agreed, added the reference [13]
- All references need to be changed, check the author's name and how they were cited in MDPI
Agreed, all references was revised and changed as recommended.
Reviewer 3 Report
This article is devoted to the isolation, characterization and study of the biological activity of Polysaccharide from South Tunisian Moringa alterniflora. The article is interesting and relevant not only for Tunisia, but also for other regions of the Mediterranean. In addition, polysaccharides play a huge role in obtaining various biologically active substances, also possessing such properties. In this regard, the authors considered this issue from different angles. Despite the abundance of advantages, there are some points that need to be finalized:
1. Why did the authors choose South Tunisian Moringa alterniflora?
2. UV–vis spectroscopy. It is known that water-soluble polysaccharides (arabinogalactans, galactomannans, glucomannans, etc.) from plant biomass do not show significant intensity in UV spectra. In addition, it is desirable to expand the description of the UV spectra.
3. Xray diffraction analysis. It is known that water-soluble polysaccharides are X-ray amorphous. This was confirmed in this work. I encourage authors to insert references to the literature on the amorphous nature of water-soluble polysaccharides. Thus, the authors can more fully describe their results.
4. Monosaccharide composition, in addition to the figure and chromatogram, must be presented in the form of a table indicating the mass content of monosaccharides. This is important because it can be used to judge branching, biological activity and some chemical and physical properties.
5. Did the authors optimize the process of obtaining the polysaccharide?
6. Are carboxyl and carbonyl groups present in the IR spectrum?
7. Why did the authors choose this particular extraction technique?
8. It is desirable to indicate what yield of water-soluble polysaccharides was after extraction.
9. Please cite: 10.3390/molecules27010266.
10. In general, more comparison of the obtained data with literature sources is needed.
11. Unification of drawings is required.
Author Response
Dear Reviewer,
We appreciate the time and effort that you and the reviewers dedicated to providing feedback on our manuscript and are grateful for the insightful comments on and valuable improvements to our paper. We have incorporated most of the suggestions made by the reviewers. Those changes are marked up using the “TrackChanges” function within the manuscript. Please see below, in green, for a point-by-point response to the reviewers’ comments and concerns.
We hope that, in the current form the MS can be accepted for publication in Plants.
Looking forward to hearing from you.
Sincerely,
Prof. Othmane Merah
- Why did the authors choose South Tunisian Moringa alterniflora?
This species wasrecently introduced in Tunisia, in the South which is an arid zone characterized by severe water deficit stress, that's why we hypothesize that the therapeutic effect could be well established in those plants.
- UV–vis spectroscopy. It isknownthat water-soluble polysaccharides (arabinogalactans, galactomannans, glucomannans, etc.) from plant biomass do not show significantintensity in UV spectra. In addition, itisdesirable to expand the description of the UV spectra.
Similar intensities (0.5-1.6) were reported in polysaccharides extracted from plants (Trabelsi et al., 2021: https://doi.org/10.1155/2021/6349019; Ben Slima et al., 2022: doi: 10.1155/2022/7858865; Ben Hsouna et al., 2022: https://doi.org/10.1002/fsn3.2836.
- X ray diffraction analysis. It is known that water-soluble polysaccharides are X-ray amorphous. This was confirmed in this work. I encourage authors to insert references to the literature on the amorphous nature of water-soluble polysaccharides. Thus, the authors can more fully describe their results.
Agreed, the following sentence was added with 2 references: "Additional data reported the amorphous nature of water-soluble polysaccharide [13,15]."
- Monosaccharide composition, in addition to the figure and chromatogram, must bepresented in the form of a table indicating the mass content of monosaccharides. This is important becauseit can beused to judgebranching, biologicalactivity and somechemical and physicalproperties.
The percentage of the monosaccharides was added in the new table 2
- Did the authorsoptimize the process of obtaining the polysaccharide?
The process of polysaccharide extraction is optimized and many MS was published by our research team: [15,16,20,35,53,57].
- Are carboxyl and carbonyl groups present in the IR spectrum?
Carboxyl group is associate with CH stretching peaks extending from 2500 to 3300 cm-1. Carbonyl stretching fall between 1900 and 1600 cm-1 which is confirmed by our results of FTIR.
- Why did the authors choose this particular extraction technique?
We choose this extraction technique because it gives the highest yield of extraction.
- It is desirable to indicate what yield of water-soluble polysaccharides wasafter extraction.
Yield results was reported in table 1
- Pleasecite: 10.3390/molecules27010266.
Agreed, this reference was added in [37]
- In general, more comparison of the obtained data with literature sources is needed.
Other data were added as recommended.
- Unification of drawings is required.
Changed as suggested
Round 2
Reviewer 1 Report
The manuscript “Polysaccharide from South Tunisian Moringa alterniflora. Lam leaves: Characterization, cytotoxicity, antioxidant activity, and laser burn wound healing in rats” has been improved and all my questions were taken into account.
I recommend the publication in “Plants”.
Author Response
Dear Reviewer
Thank youf or your valuable comments which allow us to improve our manuscript.
Reviewer 3 Report
accepted.
Author Response

(The authors gave the same response as above.)
